# Food Concepts Among Black and Hispanic Preschool-Age Children: A Preliminary Qualitative Descriptive Study Using Ethnographic Techniques and an Internet Conferencing Platform

**DOI:** 10.3390/nu17081313

**Published:** 2025-04-10

**Authors:** Celeste M. Schultz, Mary Dawn Koenig, Cynthia A. Danford

**Affiliations:** 1Department of Human Development Nursing Science, University of Illinois Chicago, Chicago, IL 60612, USA; marydh@uic.edu; 2Department of Nursing Research and Innovation, Cleveland Clinic, Cleveland, OH 44195, USA; danforc@ccf.org

**Keywords:** food concepts, preschool-age children, free lists, mouthfeel, card sort

## Abstract

**Background/Objectives**: Little is known about preschool-age children’s food concepts among diverse populations. Grounded in the Theory of Mind and Naïve Biology, the primary aim of this study was to describe Black and Hispanic preschool-age children’s food concepts. A secondary aim was to determine the feasibility of collecting data from preschool-age children via a video conferencing platform. **Methods**: Preliminary qualitative descriptive study. A purposive sample of nine 4- to 6-year-old children ($\bar{x}$ age = 4.9; Black, n = 7; Hispanic, n = 2), mostly female (n = 7) participated. Children generated two *free lists*: foods they think of, and foods they eat, reported mouthfeel of 16 foods, and performed a *constrained card sort* with rationale. **Results**: All children were able to use the video conference platform. Foods that Black and Hispanic children frequently listed as *thought of* ($\bar{x}$ = 6.75) included chicken, rice, carrots, and apples; *those* frequently listed as foods *they eat* ($\bar{x}$ = 8.33) included pancakes and grapes. Black and Hispanic children used various lexicon such as warm, soft, crunchy, and “ouchy” to describe mouthfeel. All preschool-age children sorted foods into piles (range 4–20 piles). Younger children used discrete labels to categorize foods and created many piles while older children used broader labels and created fewer piles. **Conclusions**: This is the first study to add to the literature about Black and Hispanic preschool-age children’s food concepts before receiving formal education about nutrition. Additionally, we highlight the novel and successful use of ethnographic techniques via internet video conferencing. Subtle differences in their experiential knowledge about food reflect culturally salient qualities that are critical to consider when developing interventions to promote healthy eating behavior.

## 1. Introduction

Eating behaviors of young children are influenced by a complex interplay among family context, biology, environmental and economic factors [1]. Behavior, itself, is underpinned and motivated by what people, including children, think, believe, and know as a result of their experiences with an object (e.g., food) or in a situation (e.g., eating food/mealtime) [2,3]. Outcomes to these cognitive processes, such as knowledge, are stored mentally as concepts. Translation of these assertions suggest that what children come to believe and know as a result of their experience will help to explain their current and future eating behaviors. More specifically, preschool-age children develop food concepts as they assimilate parental instruction regarding food, exposure to foods provided by their parents, observation of parental behaviors related to food, information acquired from outside sources [4] and interpretation of their sensory-motor experiences with food [5].

Young children, to some extent, are influenced by parents and primary caregivers (hereafter referred to as parent[s]) who play an active role in establishing and promoting behaviors that will likely persist throughout the child’s life [6]. Parents provide food and shape food environments by choosing which foods are available and served at mealtimes, as well as how food is presented and who eats meals together. Early-life experiences with a variety of foods offering diverse sensory-motor sensations subsequently play a role in future eating behaviors [7].

One sensory-motor sensation contributing to experiential knowledge that is often overlooked, and in which little is known about in children, is mouthfeel. Mouthfeel is an important oral or sensory experience of how food feels in an individual’s mouth [8,9,10] elicited from activation of is chemo and somatosensory pathways [9]. Mouthfeel, in part, influences food preferences [8,11]. More specifically, mouthfeel differs from texture; the sensory and functional manifestation of the structural, mechanical and surface properties of foods detected through the senses of vision, hearing, touch and kinesthetics [12], mechanical and geometric texture, as well as moisture, and fat content, or flavor (combination of taste and aroma) of food [10,13]. By four years of age, preschool-age children have had sufficient experience with food and are able to express their thoughts and understanding of their experiences. No studies are known to have examined children’s mouthfeel. Investigating mouthfeel in young children is critical to understanding its role in informing the development of food concepts that drive current eating behaviors and will likely form the foundation for future eating habits.

Preschool-age children possess a wealth of experiential knowledge related to eating; however, their knowledge is often not considered trustworthy. Children by the age of three determine that food is disgusting [14], physically contaminated, [15] socially considered contaminated [16] or dangerous [17]. Children also know what food is healthy or “junk” [18,19]. They formulate concepts and express their thoughts about their experiences with food by 36 months of age [20].

Like adults, children organize food-related concepts into categories forming a structure that speeds access to their knowledge [21]. To formulate categories, people identify common characteristics or qualities [22] that are shared among items. These characteristics or qualities often carry importance to the individual or are said to be salient [23]. Categorization underpinned by these important nuanced qualities brings order to thinking. As a result, new understanding and meaning is brought to a person’s reality, decision-making, and reasoning. Knowledge gained through experience can be more influential in directing eating behavior than knowledge of nutritional facts [24,25,26,27].

Although the literature related to young children’s knowledge of food is growing, many studies do not capture the expression of experiential knowledge directly from young children as direct informants or explicitly report the race/ethnicity of children who participated in the study [18,28,29]. Additionally, research on diverse community-dwelling young children is sparse and the use of a video conferencing platform to capture data is unknown among this population. This study further extends the science in explaining the development of eating behavior and dispels assumptions that all children have similar experiences with food.

Our study is grounded in the Theory of Mind and Theory of Naïve Biology. The Theory of Mind suggests that children, like adults, mentally organize concepts derived from their personal sensory-motor experiences to make sense of their world, and subsequently drive their behaviors (e.g., eating) [5,30]. Concurrently, the Theory of Naïve Biology focuses on young children and proposes that young children mentally construct conceptual frameworks to explain biological phenomena [31,32], such as eating before receiving formal instruction. Consequently, young children do not present as a blank slate but come with a wealth of experimental knowledge that needs to be explored and understood before effective interventions can be implemented. Therefore, our purpose was to engage Black and Hispanic preschool-age children as direct informants to describe food concepts using ethnographic techniques, and to determine the feasibility of using a video conferencing platform to elicit and collect data from preschool-age children.

## 2. Materials and Methods

### 2.1. Study Design

This preliminary qualitative descriptive study used ethnographic techniques aimed to capture Black and Hispanic preschool-age children’s concepts of food. This study was approved by a large Midwest university (RWJF IRB #2015-0353; UIC IRB 2021-0339, 7 May 2021). All parents provided written informed consent and children provided verbal assent.

### 2.2. Sample

A purposive sample of nine 4–6-year-old healthy children of color participated as direct informants. Children who were included spoke English and did not have food allergies or dietary restrictions, a diagnosis of behavioral issues (i.e., autism), endocrine disorders or syndrome, or take medications that influence appetite. Sample size was determined by the depth and quality of the data [33]. Parent’s role was to assist their child in the mechanics of using a computer or iPad to initially sort pictures of cars and trucks as a primer task and then sort foods into piles.

### 2.3. Recruitment

Preschool-age children were recruited from a daycare and a prior study of mothers with children under the age of six who agreed to participate in future research. All children resided in a large metropolitan Midwest urban area. A recruitment letter was distributed to parents of children attending a daycare and a recruitment script was emailed to mothers from the prior study.

### 2.4. Setting

This study was conducted in the child’s home via a video conference platform. The home environment provided a naturalist context where the child and parent were comfortable. Video conferencing platform was chosen as an optimal method for data collection due to in-person restrictions during the pandemic. Interviews with the children were scheduled with the parent.

### 2.5. Measures

Demographic information was collected from parents using an electronic survey instrument and included self-reported race, ethnicity, parents’ education level, and family income. 

#### 2.5.1. Interview with the Child

Semi-structured interviews were scheduled to last 30–60 min and included: (1) free lists to capture foods the child thought of and those they eat, (2) mouthfeel, (3) a primer card sort, and (4) a constrained card sort of foods (n = 73).

#### 2.5.2. Free Lists

Free lists are spontaneous listings of items, cognitive representations, or semantics assigned by groups of individuals to a category in a cultural domain [34,35]. Items within these lists are salient or carry some importance to individuals. Items listed most frequently are considered **reliable** [36]. The number of items in free lists reflect the individual’s **knowledge**; specifically, the longer the list, the more knowledgeable the individual [34,35]. Free lists have been validated for use with children across various studies [24,29,37], demonstrating their effectiveness in capturing relevant data related to food. Our process closely aligned with prior studies and was further validated through our previous research with White children.

#### 2.5.3. Mouthfeel

Mouthfeel reflects an individual’s oral sensory experience with food and contributes to an individual’s thoughts about food. As such, mouthfeel is a predictor of food consumption or refusal (i.e., eating behavior) [8,10,11]. For example, young children not only refuse food due to taste but also tactile properties [8]. Sixteen foods from the card sort with varying qualities validated by a nutritionist were paired. Paired foods included French fries and potato chips; chocolate and Skittles; apple and banana; broccoli and green beans; bread and toast; water and soda; raisins and Jell-O; crackers and applesauce. These items also appeared in the card sort.

#### 2.5.4. Primer Task for the Card Sort

A primer task is an activity administered to prepare participants for completing task(s) they will be asked to perform with the variable of interest [38]. In this study, children were asked to sort yellow, red, and green cars and trucks into piles and tell their reasons for sorting the vehicles. These instructions mirrored the instructions for the card sort with foods.

#### 2.5.5. Card Sort

Card sort is an ethnographic technique believed to reflect how people organize their thoughts (i.e., cognitive constructs around a domain) [39]. When this technique is used, participants are asked to sort images into piles. Sorting can be constrained (e.g., limited number of piles; ranking), or unconstrained (e.g., no limitations). Children were asked to sort single images of food items “any way they wanted” and “explain their reasoning for sorting”. Card sorts have also been validated for use with children in prior studies [24,37]. The number of possible piles was constrained to 20 due to software limitations. Foods for the card sort were developed from previous studies that used pictures to determine how children categorized food [18,28]. Images of foods were obtained from the internet and commercial food products.

### 2.6. Procedure

A video conferencing platform was selected for data collection due to in-person restrictions during the COVID-19 pandemic. After parents logged into the video conference platform, children were positioned on the parent’s lap or on a chair in front of the computer screen with their parent behind them. Parents were instructed to limit their guidance and allow their child to respond and perform tasks independently. Children were told that the researcher was interested in knowing what they knew about food. First, children participated in verbally generating free lists of foods they could think of and those they eat. Next, children were asked to report how 16 contrasting foods felt in their mouth. Children were then asked to perform a primer task sorting cars and trucks to prepare for sorting foods in the card sort. Finally, children were asked to perform the constrained card sort with food. Upon completion of the study, parents and children collectively were compensated with a gift card from Walmart worth twenty dollars.

### 2.7. Analysis

Descriptive statistics were used to analyze demographic data. Frequency of foods listed in free lists were calculated. Content analysis was used to gain a rich understanding of mouthfeel and card sorts. Dependability (reliability) and confirmability (validity) were assured through a systematic and rigorous process of data collection and data analysis, comparison of findings to prior research, audit trails, and methodological and investigator triangulation. All three authors simultaneously reviewed the video recordings and transcripts. In-depth rigorous discussion among the authors followed to achieve agreement on the interpretation of the data.

## 3. Results

### 3.1. Demographic

Seven Black children and two Hispanic children ages four (n = 6), five (n = 1), or six years old (n = 2) participated. Children were primarily female (sex assigned at birth); two 4-year-old males (sex assigned at birth) participated. Parents were primarily mothers; one was the child’s grandmother. All parents were the same race as the child. Parents reported their highest education as high school (n = 3), some college or technical school (n = 2), completed college education (n = 3), or a graduate degree (n = 1). Most family incomes ranged from $60,000 to 149,999 (n = 5) with one greater than $150,000, and three less than $60,000.

### 3.2. Interview Using Video Conferencing Platform

All children completed the interview via a video conferencing platform. The interview lasted on average 39.59 min (range 9.59 to 54.45 min). Some interviews varied in duration as some children quickly accommodated to the logistics of the online process and completed the task quickly. For others, interviews were longer as their attention waned, or they became fatigued. Occasionally, a child needed a break to use the bathroom or get a snack. One child was active and needed intermittent prompts to refocus.

### 3.3. Free Lists

#### 3.3.1. Foods Thought of

Eight children completed the free lists of foods they thought of over an average of 1.18 min (range 0.20 to 1.43 min). One four-year-old child did not complete the free lists due to being unfocused and needing the visual nature of the mouthfeel and card sort tasks to maintain their attention. Children listed 2 to 11 foods they thought of ($\bar{x}$ = 6.83; median = 7), totaling 30 different foods (Table 1).

#### 3.3.2. Foods They Eat

Six children completed the free lists of food they eat over an average of 1.26 min (range 0.04 to 2.25 min). Three four-year-old children did not complete the list. Children listed 5 to 13 foods they eat ($\bar{x}$ = 8.33; median = 5), totaling 31 different foods (Table 2).

#### 3.3.3. Comparison Between Foods in Both Free Lists

A total of 39 different foods were listed between both free lists. Of the foods children thought of (n = 36), 13 were listed as foods they eat.

### 3.4. Mouthfeel

All the children reported how food felt in their mouth. Interviews addressing mouthfeel lasted an average of 5.38 min (range 4.22 to 7.66 min). Nearly half of the children reported that French fries felt good (n = 4) while one-third reported French fries felt soft. One-third of children reported that potato chips felt crunchy and good. One-third of children reported that chocolate and Skittles felt good. Children (n = 3) reported that apples felt crunchy while bananas felt good (n = 4) and soft (n = 3). Children (n = 2) reported that broccoli felt hard and crunchy and green beans felt soft and good. Five children reported that bread felt soft while two children reported toast felt hard, hot, and soft. Nearly one-half of children described water as cold (n = 4) and one-third of children (n = 3) reported that water felt wet. Two children reported that soda felt good. Children (n = 2) reported that raisins felt good and squishy, and Jell-O felt good (n = 2). Children reported that crackers felt crunchy (n = 3) and hard (n = 4) while applesauce felt soft (n = 4).

### 3.5. Primer Task

Eight children completed the primer over an average of 3.06 min (range 0.97 to 5.37 min). One child did not complete the primer card sort to maintain their attention. Children sorted red, green, and yellow trucks and cars based on color (n = 1), vehicle type (n = 3), or mixed characteristics (some piles were vehicles, other piles were color) (n = 4).

### 3.6. Card Sort

All children completed the card sort of foods. Children completed the card sort over an average of 23.28 min (range 7.67 to 36.27 min) and spontaneously named the foods as they sorted the images. The number of piles ranged from four to twenty ($\bar{x}$ = 16.4). Four children sorted all the foods; five children sorted a portion of the foods. Children categorized foods by various labels (see Table 3).

## 4. Discussion

This is the first manuscript known to describe Black and Hispanic preschool-age children’s concepts of food. Previous studies exploring children’s food concepts have included primarily White preschool-age children from university daycare settings [18,24,40,41,42,43]; only two studies are known to have included Black and Hispanic 5- to 11-year-old children recruited from urban and suburban settings [40,41]. However, data from these studies were not analyzed by ethnicity and race. Unlike prior studies, our participants were recruited from diverse community-dwelling populations. Additionally, in prior studies, children were asked to sort foods using predetermined categories, whereas in our study children were asked to categorize foods in their own way to reflect their understanding of food. Our study is also the first known to explore mouthfeel and to conduct an interview via a video conference platform with preschool-age children.

### 4.1. Interview Techniques with Preschool Children

We used a developmentally appropriate approach incorporating unique techniques of free lists, card sorts and mouthfeel to elicit data during interviews with the children. The free list and card sort techniques were successfully used in our similar study with White preschool children [37]. In this study, Black and Hispanic children were equally as engaged in talking about the foods they know and how they categorize foods. Mouthfeel was included to gain insight into preschool-age children’s thoughts about food resulting from their oral experiences.

#### 4.1.1. Foods Preschool-Age Children Think About and Eat

Using free lists, most Black and Hispanic preschool-age children in our study spontaneously generated lists of foods demonstrating that they have experiential knowledge about food even before receiving structured instruction. Our results were similar to other studies of school-age children [44]. The average number of items identified in free lists, also known as list length, were similar to the list lengths generated by preschool-age White children who were obese and of healthy weight [37]. This finding suggests that children from various racial and ethnic backgrounds share a similar breadth of knowledge of food [45] even though they may report different foods. Some preschool-age children in our study did not complete the free list, which may be attributed to their developmental maturity or unfamiliarity with some of the foods or use of the video platform and technology.

Interestingly, Black and Hispanic preschool-age children in our study were more knowledgeable about the foods they eat than those they could think of. This finding is unlike the knowledge of preschool-age White children who were obese and of healthy weight who listed more foods they thought of than those which they ate [37]. Our results may indicate that parental emphasis about eating food or the child’s exposure to food is more salient than an emphasis on knowledge about food (e.g., healthy or unhealthy).

Unlike studies using similar techniques with school-age children, our study found subtle differences in the foods our preschool-age children thought of and those they eat. Like school-age children in studies from the Midwest United States, Argentina and France, children in our study listed apples, bananas, strawberries, grapes, and oranges [37,46], and carrots and broccoli [37,44]. Additionally, the preschool-age children in our study listed meat, fish, and eggs. Differences in foods listed across the studies may be attributed to study methodology. For example, adolescents and school-age children in Argentina and France were asked to list foods by a specific predetermined category (e.g., fruits or vegetables) versus being asked to spontaneously list foods without taxonomy categorization prompts [44,46]. Listing of meat, fish, and eggs may indicate that those foods are in some way important and familiar to children in our study [34,35]. Differences in these food lists suggest that important cultural differences may exist in the food domain among diverse populations [47]. Investigating food and eating behaviors needs to be tailored to culturally salient practices and needs to be interpreted with caution when determining if dietary practices are healthy.

#### 4.1.2. Interpretation of the Feel of Food in Preschool-Age Children’s Mouths

The preschool-age children in our study were able to describe the conceptual representation of their oral sensation (how foods feel in their mouth), which is different than describing food texture, the physical property of food (e.g., fatty, moist). Although mouthfeel and texture are sometimes used interchangeably, young children in this study were able to distinguish between mouthfeel and texture.

No other studies are known to have investigated mouthfeel with children. In our study, oral sensation of food described by preschool-age children formed the basis and helped to inform the development of food concepts. Their concepts were represented by various words (e.g., wiggly, soft, and warm) also known as linguistic synthesis [48]. To further illustrate, children described soda as “stingy”. Notably, adults also assigned the term stingy to soda. Similar use of terminology by adults and children when describing soda is representative of the mounting evidence that the oral sensation of drinking soda is attributed to activation of nociceptive nerve endings in the oral mucosa from carbonic acid in the beverage [9,49].

In our study, Black and Hispanic preschool-age children also described their sensation as “good”. The meaning of the word “good” is vague and may have many meanings for young children. Good may imply that the food is “just right” [48]. “Good” may also mean that the food feels good, tastes good, is good for you, or that the children simply like the food. Because mouthfeel is also associated with food preference and moderately to strongly related to food intake [8,10,11], future studies are needed to explore the meaning of “good” and should include identification of foods children prefer, like or dislike, and those they eat.

#### 4.1.3. Card Sorts

Prior research shows that children by four years of age have the cognitive ability to categorize items based on subtle qualities or characteristics assigned by groups of individuals within their social or cultural environment [50,51,52,53]. Vygotsky further elucidated that “concepts emerge from children thinking about their daily experiences which occur in the context of normal participation in family and community practices and activities”, also known as social norms [54]. Children in our study sorted foods into various piles to reflect their organization of concepts (thoughts) around food. These concepts were reflected in the labels they assigned to the piles. For example, like 3-year-old children in previous studies, children in our study classified food as “healthy” or “junk”: an *evaluative category* [18,19]. Like school-age children (5- and 6-year-olds) [41], preschool-age children in our study also categorized foods as “sweet”, or as “meat”: a food category or *taxonomy*.

Children in our study further associated foods in the card sort with meals, such as “breakfast”, known as *script categorization*, or with properties associated with mouthfeel, such as hard, soft, or cold. Use of terms designating evaluation, taxonomy, or script by preschool-age children aligns with categories used by undergraduate students to classify foods in a seminal study [28]. Identification of these food properties also parallels those found in our previous study among White preschool-age children with obesity [37] and Pickard et al.’s (2023) study [55]. Interestingly, children in our present study did not categorize individual food items as vegetables even though they listed them as a food they thought of and those they eat. Reasons for not categorizing food items in the card sort as vegetables is unclear as children as young as infants have the capacity to infer categories based on sensory-motor experience, perception, and abstraction [56,57]. Some may argue that this lack of individual identification of vegetables may be attributed to language used at mealtime, such as “eat your vegetables”. However, others argue that this lack of categorization of individual vegetables cannot be attributed to children mimicking their parents’ linguistics [57]. Still others could argue that the lack of vegetable categorization could be due to parental nutrition knowledge or education [58], or exposure to a variety of foods due to accessibility, availability, and affordability [59]. Therefore, further investigation of this phenomenon is warranted.

Lastly, 4- and 5-year-olds used multiple discrete labels while 6-year-olds used broad concepts to categorize foods. These findings may be explained by the developmental progression of conceptualization from concrete to abstract among these age groups [50,56].

Variations in food concepts seen in our study and others could be due to social and family-specific assigned characterization of food. Evidence shows that parents rely on their own experiences or knowledge gained from familial, cultural, and/or social groups, as well as their own personal history with foods, to conceptualize their understanding of food and frame their eating behaviors [60,61]. Moreover, parental nutrition knowledge and education level directly influence children’s knowledge of healthy and unhealthy foods [43]. Dissimilarities across studies also give rise to the question of whether differences exists in what children are being told about food (instructional), what they observe (vicarious), and/or what they personally (sensory-motor) experience with food [49,62,63]. Therefore, societal and familial emphasis on knowledge of or behavior associated with food (e.g., eating) needs to be considered in future research.

### 4.2. Strengths and Limitations

This is the first known study to describe the foods that Black and Hispanic preschool-age children *think of* and *eat*, how food feels in their mouth, and how these children categorize food. Our approach is unique as it demonstrates that children can spontaneously assign categories to food rather than sort foods into predetermined categories. Our approach provides greater insight into what children know about food before they receive formal nutritional instruction. We speculate that eating behavior may be more closely linked to these “naïve” concepts rather than formal nutritional knowledge, as little or no association between formal nutritional knowledge of food and eating behaviors has been found [24,25,26,27]. Additionally, we demonstrated success in collecting data from preschool-age children using ethnographic techniques and a virtual platform.

Although cognitive readiness was not determined, all children completed the interview and nearly all the tasks. Despite instructions requesting that they not guide the child, parents occasionally rephrased the original question about food and prompted their child to distinguish if these were their favorite foods or those they eat. Lack of familiarity with the use of technology or short attention span may have impeded some children’s ability to complete all tasks using the virtual platform. Potential bias toward those with computer technology was considered; however, nearly 97% of the population in the United States have computer access (93%) or at least a smart phone (4%) [64]. Evidence also shows increasing frequency of computer use in preschools [65]. Lastly, although qualitative findings are transferable, caution must be used in applying this evidence to the general population. Qualitative results can only be generalized after several replications.

### 4.3. Future Research

Enlisting parents or significant caregivers to identify and categorize foods in future studies along with their child may provide more insightful evidence in explaining children’s thoughts about food. This approach will allow researchers to look at how parents and children are similar or different. In addition, more data are needed on location of food consumption (i.e., restaurant, home, etc.), as well as who the child eats with and what they eat. This data may be helpful in revealing the origin of children’s thoughts about food. Collecting data on the children’s and parents’ weight may also be helpful in explaining thought processes surrounding food with eating behaviors.

## 5. Conclusions

Our findings not only add to the evidence showing that children can and do formulate concepts of food by the age of four [18,19,24,66,67], but also demonstrates subtle yet important differences in the foods our participants listed and how foods were categorized. Black and Hispanic children were most familiar with carrots, rice, chicken, grapes, and pancakes, and most frequently formulated categories as foods that “go together”, and meat. Evidence from this and previous studies suggest that some conceptual similarities and differences exist between diverse populations regarding food. More importantly, understanding nuanced differences may be crucial in developing effective, sustainable interventions to address foods to eat and eating behaviors. More specifically, health care providers need to explore experiential knowledge about food with the child and family in order to make meaningful recommendations.

## Figures and Tables

**Table 1 nutrients-17-01313-t001:** Foods that Children *Thought of*.

Food Group	Frequency	4 Years (n = 6)	5 Years (n = 1)	6 Years (n = 2)
Veggies	1 time	broccoli, cucumbers, pickles, peas, cabbage, salad	vegetables	broccoli, carrots
	>1 time	carrots (2)	-	-
Fruits	1 time	-	apples, oranges	apples, oranges grapes, lemon
Meats/Fish/Eggs	1 time	fish sticks, hot dog, fish, hamburger, eggs, bacon	chicken	-
	>1 time	chicken (2)	-	-
Grains/Breads	1 time	waffle, noodles, oatmeal, pasta	rice, cereal, macaroni	-
	>1 time	rice (2), cereal (2)	-	-
dairy	1 time	-	-	-
Combo Foods	1 time	pizza	-	-
	>1 time	spaghetti (2)	-	-
Sweets/Junk Food	1 time	cupcakes	-	-
Drinks	1 time	-	-	-

**Table 2 nutrients-17-01313-t002:** Foods that Children *Eat*.

Food Group	Frequency	4 Years (n = 6)	5 Years (n = 1)	6 Years (n = 2)
Veggies	1 time	broccoli, peas, cabbage, vegetables	-	broccoli, carrots
	>1 time	carrots (2)	-	
Fruits	1 time	apples, banana, juice, blueberries	-	grapes, strawberries
	>1 time	grapes (3), oranges (2), strawberries (2), pineapple (2)	-	apples (2)
Meats/Fish/Eggs	1 time	chicken nuggets, beans, chicken, fish	chicken	-
Grains/Breads	1 time	French toast, pancakes, toast, crackers	pancake, rice	French toast
	>1 time	-	-	pancakes (2)
dairy	time	ice cream, yogurt	-	-
Combo Foods	1 time	-	spaghetti	pizza, corn dogs
	>1 time	spaghetti (2)	-	-
Sweets/Junk Food	1 time	Doritos	jellybeans	-
	>1 time	chocolate (2)	-	-
Drinks	1 time	-	-	water

**Table 3 nutrients-17-01313-t003:** Categorization of foods in card sort (N = 9).

Category (Frequency)	4-Year-Old (n = 6)	5-Year-Old (n = 1)	6-Year-Old (n = 2)
Descriptors	Eat that food together (5)	Go together (1)	Foods that go together (3)
	Health (4), Cold (3), Hard (3), Junk (2), Soft (2), Salty (2)	-	-
	Sweet (2)	Sweet (1)	-
	-	Favorite (2), Crunch (1), Disgusting (1), Like (1), So good (1)	-
Food Classification	Meat (2)	-	Meat (1)
	Fruit (2)	-	
	-	-	Bread (1)
Food type	Candy (2), Sandwich (2), Noodle (2), Chicken (1), Bacon (1), Milk (1), Corn (1)	-	-
	-	Drink (soda) (1)	-
Meal type	Breakfast (2), Snack (1)	-	-

## Data Availability

We are unable to release the original data as it is video recorded, and children and their mother’s faces are visible. Release of the original data would jeopardize the anonymity and confidentiality of the participants.

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
