# Peer review of "Food Concepts Among Black and Hispanic Preschool-Age Children: A Preliminary Qualitative Descriptive Study Using Ethnographic Techniques and an Internet Conferencing Platform"

_nutrients, 2025, doi:10.3390/nu17081313_

Round 1

Reviewer 1 Report

Comments and Suggestions for Authors

The paper “Food Concepts Among Black and Hispanic Preschool-age Children: Qualitative Descriptive Research” addresses the subject of nutritional habits in children and should contribute to the growth of the literature on this research (especially for nutritionists and economists) and food producers for children.

Before  the manuscript acceptation for publication in “Nutrients”, the following items should be revised:

The title

The analysis is based on a small sample of children, which is not representative of the study's sample.

Isn't the aim too general? For "Black and Hispanic children " n=7?

Can hypotheses be considered with such a group?

"as primary participants" - were other children studied?

Therefore, our purpose was to engage Black and Hispanic preschool-age children as primary participants to describe food concepts using ethnographic techniques, and to determine the feasibility of using a video conferencing platform to elicit and collect data from preschool-

age children. We hypothesized that Black and Hispanic children would complete ethnographic tasks to reveal their food concepts, and would complete the study using a video conferencing platform"

Were other children studied?

Studies among children are difficult. How were the questions validated? - Were they understood in the same way by all the children?

Line 128-129

"Foods for the card sort were developed from previous studies that used pictures to determine how children categorized food [16,26]. "- these studies were conducted on a larger group of children.

Results:

There is no statistical significance of these results

Author Response

Comments and Responses 1 Round 1

The paper “Food Concepts Among Black and Hispanic Preschool-age Children: Qualitative Descriptive Research” addresses the subject of nutritional habits in children and should contribute to the growth of the literature on this research (especially for nutritionists and economists) and food producers for children.

Before  the manuscript acceptation for publication in “Nutrients”, the following items should be revised:

  1. Comment: The title

Response: Although the concern for the title is unclear, we have revised the title consistent with reviewer 2. The title now reads: Food Concepts Among Black and Hispanic Preschool-age Children: A Qualitative Descriptive Study Using Ethnographic Techniques and an Internet Conferencing Platform

  1. Comment: The analysis is based on a small sample of children, which is not representative of the study's sample.

Response: Thank you for your comment. Although the sample size may be considered small, the intent of qualitative research is to gain insight into the participant’s concepts; it is the depth of content that is priority not the sample size. In addition, the sample size is like those of other qualitative studies with children.

  1. Comment: Isn't the aim too general? For "Black and Hispanic children " n=7?

Response: Thank you for your comment. As this is the first study to collect data from Black and Hispanic. children (n = 9), our goal was to gain insight into their concepts using ethnographic techniques and test the feasibility of using an internet conferencing platform to collect data.

  1. Comment: Can hypotheses be considered with such a group?

Response: Thank you for your comment. On further review, we have deleted the sentence as the intent of this study was not to be predictive.

  1. Comment: "as primary participants" - were other children studied?

Response: Thank you for your comment. The primary participants in our study were children. Studies involving children often obtain data from parents to explain their behavior. Data from parents, however, may not truly reflect children’s knowledge and understanding that underpins their behavior. Our study collects data directly from children. This approach substantiates children’s cognitive abilities and reveals accurate drivers to their behavior.

  1. Comment: Therefore, our purpose was to engage Black and Hispanic preschool-age children as primary participants to describe food concepts using ethnographic techniques, and to determine the feasibility of using a video conferencing platform to elicit and collect data from preschool- age children. We hypothesized that Black and Hispanic children would complete ethnographic tasks to reveal their food concepts, and would complete the study using a video conferencing platform"

Response: Thank you for your comment. We are uncertain the information or revision you are seeking. As described above, we deleted the sentence (we hypothesized) to eliminate confusion and redundancy.

  1. Comment: Were other children studied?

Response: Thank you for your comment. We are uncertain regarding the intent of this question. There were no other children included in this study. Our sample in this study consisted of 9 preschool-age children. Our prior study included 30 preschool-age children who were obese and 30 preschool-age children who were healthy weight.

  1. Comment: Studies among children are difficult. How were the questions validated? - Were they understood in the same way by all the children?

Response: Thank you for your comment. We have added clarity to the free list and card sort measures: Free lists have been validated for use with children in a variety of studies (Schultz & Danford, 2016; Schultz & Danford, 2021; Nguyen & Murphy, 2003). Children were given consistent instructions. Based on children’s performance, 8 of the 9 children completed the task without difficulty. Card sorts have also been validated for use with children (Schultz & Danford, 2016; Schultz & Danford, 2021). Based on the fact that all children completed the card sort would suggest that children understood the questions/directions.

Reviewer 2 Report

Comments and Suggestions for Authors

General Assessment
The study examines food concepts among Black and Hispanic preschool-age children using qualitative ethnographic techniques and evaluates the feasibility of video conferencing for data collection. The study is relevant, but there are areas that require clarification and further elaboration.

Major Comments (With Line Numbers)
1. Title & Abstract
Line 2-3: The title is clear but could be more specific. Consider emphasizing the qualitative methodology and video conferencing aspect.
Line 9-12: The abstract lacks a brief mention of theoretical frameworks. Consider adding how the study builds on existing literature.
Line 26-28: The conclusion is well-written, but it could highlight the novelty of the video conferencing approach more explicitly.
2. Introduction
Line 37-41: The discussion of cognitive processes is useful, but there is a need to cite more specific literature on how preschool-age children develop food concepts.
Line 50-55: The explanation of mouthfeel is thorough, but it could be more concise. Consider integrating references more smoothly rather than listing multiple citations back-to-back.
Line 68-71: The justification for focusing on Black and Hispanic children is strong, but the authors should clarify how this study extends previous research beyond just population diversity.
3. Methods
Line 79-81: The choice of ethnographic techniques is appropriate, but the authors should clarify why a qualitative descriptive study was chosen instead of an alternative qualitative approach (e.g., phenomenology, grounded theory).
Line 83-86: The exclusion criteria related to dietary restrictions and behavioral issues should be better justified. How might excluding children with dietary restrictions influence the findings?
Line 96-98: The decision to use a video conferencing platform should include a discussion of limitations (e.g., potential technological barriers).
Line 106-109: The free list method is well explained, but how was reliability ensured? Was any inter-rater agreement assessed?
Line 121-126: The constrained card sort should have included more details about how the categories were determined. Were children given any instructions on potential categorization criteria?
Line 132-137: How were potential distractions (e.g., parental influence, home environment) controlled? This could introduce bias in the results.
4. Results
Line 157-161: The discussion of interview length is insightful, but it would be helpful to include a statistical measure of variance (e.g., standard deviation) to supplement the range.
Line 165-167: One child did not complete the free list task—did this affect the results? Were any adjustments made to account for missing data?
Line 188-189: The observation that children described water as "wet" and "cold" is interesting. However, did they exhibit any understanding of abstract versus concrete descriptors?
Line 198-200: The number of food piles varied widely (4-20). Was there any correlation between age and number of piles? This could help interpret developmental trends.
5. Discussion
Line 209-215: The comparison to previous studies is well done, but the authors should discuss potential limitations in cultural generalizability.
Line 226-229: The claim that Black and Hispanic children have experiential knowledge of food before formal education is valuable. However, more discussion on how this compares with White children in previous studies would be helpful.
Line 260-262: The discussion of soda being described as “stingy” is insightful. The authors could further elaborate on how this aligns with adult perceptions of carbonation.
Line 275-278: The discussion on categorization could be expanded by including more references to cognitive development research (e.g., Piaget, Vygotsky).
Line 287-289: The lack of vegetable categorization is intriguing. Could this be due to parental influence or social norms?
Line 295-297: The discussion on parental influences is important. The authors should consider whether parental nutrition education level influenced children’s responses.
6. Conclusion
Line 329-333: The conclusion effectively summarizes key findings, but it would be stronger if it explicitly stated the implications for future research and interventions.
Line 336-337: The suggestion for future interventions is useful but should include practical strategies for implementation.

Author Response

Comments and Responses 2 Round 1

The study examines food concepts among Black and Hispanic preschool-age children using qualitative ethnographic techniques and evaluates the feasibility of video conferencing for data collection. The study is relevant, but there are areas that require clarification and further elaboration.

  1. Comment: Title & Abstract Line 2-3: The title is clear but could be more specific. Consider emphasizing the qualitative methodology and video conferencing aspect.

Response: Thank you. We have modified our title to state: Food Concepts Among Black and Hispanic Preschool-age Children: A Qualitative Descriptive Study Using Ethnographic Techniques and an Internet Conferencing Platform

  1. Comment: Title & Abstract Line 9-12: The abstract lacks a brief mention of theoretical frameworks. Consider adding how the study builds on existing literature.

Response: Thank you. We have added the theoretical frameworks to the abstract second sentence in the background: Grounded in the Theory of Mind, and Naïve Biology, the primary aim was to…

We have added a brief discussion of the theoretical frameworks to the background prior to the methods section.

Bartsch, K., & Wellman, H. M. (1989). Young children’s attribution of action to beliefs and desires. Child Development, 60(4), 946-964.

Wellman, H. M., & Bartsch, K. (1998). Young children’s reasoning about beliefs. Cognition, 30, 239-277.

Inagaki, K., & Hatano, G. (1990). Young children’s knowledge in everyday biology. British Journal of Developmental Psychology, 8, 281-288.

Inagaki, K., & Hatano, G. (2006). Young children’s conception of the biological world. Current Directions in Psychological Science, 15, 177-181.

We have also addressed your concern regarding how this study adds to the literature. This sentence replaces the first sentence in the conclusion: This is the first study to add to the literature about Black and Hispanic preschool-age children’s food concepts before receiving formal education about nutrition.

  1. Comment: Title & Abstract Line 26-28: The conclusion is well-written, but it could highlight the novelty of the video conferencing approach more explicitly.

Response: Thank you for your comment. Additionally, we highlight the novel and successful use of ethnographic techniques via video conferencing.

  1. Comment: Introduction Line 37-41: The discussion of cognitive processes is useful, but there is a need to cite more specific literature on how preschool-age children develop food concepts.

Response: Thank you for your comment. We have included a sentence to further illustrate the development of food concepts: More specifically, preschool-age children develop food concepts as they assimilate parental instruction regarding food, exposure to foods provided by their parents, observation of parental behaviors related to food, information acquired from outside sources, and interpretation of their sensory motor experiences with food. McCaffee, 2003, McCafee, J. (2003). Childhood eating patterns: The role parents play. Journal of the American Dietetic Association, 103(12), 1587. Doi: 10.1016/j.jada.2003.10.031

  1. Comment: Introduction Line 50-55: The explanation of mouthfeel is thorough, but it could be more concise. Consider integrating references more smoothly rather than listing multiple citations back-to-back.

Response: Thank you for your comment. Reviewer three requested further clarification regarding mouthfeel. We added the following sentence to the paragraph: By four years of age, preschool-age children have had sufficient experience with food and are able to express their thoughts and understanding of their experience. No studies are known to have examined children’s mouthfeel. Investigating mouthfeel in young children is critical to understanding its role in informing the development of food concepts that drive current eating behaviors and will likely form the foundation for future eating habits. 

  1. Comment: Introduction Line 68-71: The justification for focusing on Black and Hispanic children is strong, but the authors should clarify how this study extends previous research beyond just population diversity.

Response: Thank you for your comment. We have added one sentence to add clarity regarding our contribution to science: This study further extends the science in explaining the development of eating behavior and dispels assumptions that all children have similar experiences with food.

  1. Comment: Methods Line 79-81: The choice of ethnographic techniques is appropriate, but the authors should clarify why a qualitative descriptive study was chosen instead of an alternative qualitative approach (e.g., phenomenology, grounded theory).

Response: Thank you for your comment. Qualitative descriptive approach best fits our purpose in describing children’s food concepts and explaining the feasibility of using the video conferencing platform. The intent was not to deeply explain a phenomenon or build theory. – Sandolowski , 2000. Sandelowski, M. (2000). What ever happened to qualitative description? Research in Nursing & Health, 23, 334-340. vs phenomenology – study of a phenomenon – Vs grounded theory Glaser, B. G., & Straus, A. L. (1967). The Discovery of Grounded Theory: Strategies for Qualitative Research.

  1. Comment: Methods Line 83-86: The exclusion criteria related to dietary restrictions and behavioral issues should be better justified. How might excluding children with dietary restrictions influence the findings?

Response: Thank you for your comment. We desired to have children that had no limitations in their diet so that consistency was maintained. as not to introduce confounding factors. We added a phrase to the end of the second sentence: to establish baseline data.

  1. Comment: Methods Line 96-98: The decision to use a video conferencing platform should include a discussion of limitations (e.g., potential technological barriers).

Response: Thank you for your comment. We address this issue in the limitation section of this manuscript.

  1. Comment: Methods Line 106-109: The free list method is well explained, but how was reliability ensured? Was any inter-rater agreement assessed?

Response: Thank you for your comment. Reliability in free lists is assured based on the frequency by which items are listed. This is included in text. Reliability is not appropriate for qualitative studies; however, transferability is addressed later in the manuscript.

Inter-rater agreement is not appropriate for free lists.

  1. Comment: Methods Line 121-126: The constrained card sort should have included more details about how the categories were determined. Were children given any instructions on potential categorization criteria?

Response: Thank you for your comment. Children spontaneously assigned labels or explained their rationale for sorting, indicating categorization of the food items. Children were not given examples of categories to allow for free expression of their knowledge and avoid bias. We clarify this concern by adding quotation marks around “any way they wanted” and added “explain their reasoning for sorting.” Children had an opportunity to practice categorization through the primer task.

  1. Comment: Methods Line 132-137: How were potential distractions (e.g., parental influence, home environment) controlled? This could introduce bias in the results.

Response: Thank you for your comment. We added further clarification on parental instruction in section 2.6 procedure.

  1. Comment: Results Line 157-161: The discussion of interview length is insightful, but it would be helpful to include a statistical measure of variance (e.g., standard deviation) to supplement the range.

Response: We appreciate your comment. This is a qualitative study. As such, it does not focus on numerical variance but rather examines how responses differ. We explain the variance in the length of interviews based on children’s behavior rather than numerically. Therefore, we believe inclusion of standard deviation would not be appropriate.

  1. Comment: Results Line 165-167: One child did not complete the free list task—did this affect the results? Were any adjustments made to account for missing data?

Response: Thank you for your comment. The results were not affected by one child not completing the free list. The data were analyzed based on the remaining 8 children. No adjustment was made for missing data as this is a qualitative study.

  1. Comment: Results Line 188-189: The observation that children described water as "wet" and "cold" is interesting. However, did they exhibit any understanding of abstract versus concrete descriptors?

Response: We appreciate your comment. Observation of children’s behaviors related to their understanding of lexicon was beyond the scope of this study.

  1. Comment: Results Line 198-200: The number of food piles varied widely (4-20). Was there any correlation between age and number of piles? This could help interpret developmental trends.

Responses: Thank you for your comment. This is indirectly addressed in the categories – younger more (1 to 7) granular categories than older children who used broader categories (1).  In addition, the sample size does not allow for the calculation of correlations and is inconsistent with qualitive research.

  1. Comment: Discussion Line 209-215: The comparison to previous studies is well done, but the authors should discuss potential limitations in cultural generalizability.

Response: Thank you for your comment. Findings from qualitative work are not generalizable, only transferable. At this point, studying preschool-age children’s food concepts has not been well studied. Therefore, our study provides new insight into salient differences in cultural influence.

  1. Comment: Discussion Line 226-229: The claim that Black and Hispanic children have experiential knowledge of food before formal education is valuable. However, more discussion on how this compares with White children in previous studies would be helpful.

Response: This is an interesting comment. This is beyond the scope of this study, as this is the first with Black and Hispanic children. We do state under the free lists that children from different racial and ethnic backgrounds share similar breadth of knowledge of food compared to White children but they list different items.

  1. Comment: Discussion Line 260-262: The discussion of soda being described as “stingy” is insightful. The authors could further elaborate on how this aligns with adult perceptions of carbonation.

Response: Thank you for your comment. To add clarity, we added a sentence to state: Notably, adults also assigned the terms stingy to soda.

  1. Comment: Discussion Line 275-278: The discussion on categorization could be expanded by including more references to cognitive development research (e.g., Piaget, Vygotsky).

Response: Thank you for your comment. We have added two classic references – 1. Gelman, S. & Kalish, C. W. (2008). In Child and Adolescent Development An Advanced Course, Damon, W. Lerner, R. M. Eds., Conceptual Development, 298-321. Wiley & Sons, Inc.   2. Markman, E. M., Cox, B., & Machida, S. (1981). The standard object-sorting task as a measure of conceptual organization. Developmental Psychology, 17(1), 115-117. https://doi.org/10.1037/0012-1649.17.1.115

  1. Comment: Discussion Line 287-289: The lack of vegetable categorization is intriguing. Could this be due to parental influence or social norms?

Response: Thank you for your comment. We added also known as social norms to line 294. We also added: Still others could argue the lack of vegetable categorization could be due to parental nutrition knowledge or education (Vereecken, c., Maes, L. (2010). Young children’s dietary habits and associations with the mothers’ nutritional knowledge and attitudes. Appetite, 54, 44-51. https://doi.10.16/j.appet.2009.09.005 or exposure to a variety of foods due to accessibility, availability, and affordability (Birch, 1999). Birch, L. L. (1999). Development of food preferences. Annual Review of Nutrition, 19, 41-62.

  1. Comment: Discussion Line 295-297: The discussion on parental influences is important. The authors should consider whether parental nutrition education level influenced children’s responses.

Response: Thank you for your comment. We have added a sentence indicating: Moreover, parental nutrition knowledge and educational level also directly influence children’s knowledge of healthy and unhealth foods. Zarnowiecki, D., Sinn, N., Petkov, J., Dollman, J. (2011). Parental nutrition knowledge and attitudes as predictors of 5-6-year-old children’s healthy food knowledge. Public Health Nutrition, 15(7), 1284-1290. https://doi.10.107/S136898001103259

  1. Comment: Conclusion Line 329-333: The conclusion effectively summarizes key findings, but it would be stronger if it explicitly stated the implications for future research and interventions.

Response: Thank you for your comment. We address this under section 4.3 Future research

  1. Comment: Conclusion Line 336-337: The suggestion for future interventions is useful but should include practical strategies for implementation

Response: Thank you for your comment. We have added a sentence to address your comment: More specifically, health care providers need to explore the experiential knowledge about food with the child and family in order to making meaningful recommendations.

Reviewer 3 Report

Comments and Suggestions for Authors

Dear authors,

Your article titled "Food Concepts Among Black and Hispanic Preschool-age Children: Qualitative Descriptive Research" addresses a relevant and underexplored topic concerning the food concepts of Black and Hispanic preschool-aged children, employing qualitative descriptive research. The use of ethnographic techniques and a video conferencing platform during the pandemic is innovative and appropriate. The findings provide important insights into how these children conceptualize food before formal nutrition education. 

Please consider some comments: 

1.  The sample size of nine children is small, which limits the generalizability of the findings. While qualitative research often uses small samples, a more detailed justification for this sample size would strengthen the article. Additionally, including a brief discussion on how the sample size impacts the transferability of the results would be beneficial.

2. While the concept of mouthfeel is introduced well, the distinction between mouthfeel and texture could be further clarified, especially in the context of preschool children’s understanding. Adding more context about why this distinction is critical for this age group would improve comprehension.

3. The article would benefit from more in-depth analysis of how the children’s food concepts differ from other ethnic groups or age groups. While there is a comparison to White preschool children in prior studies, a more structured comparison of key similarities and differences in the results could highlight the study’s contributions more effectively.

4.  The article mentions that parents occasionally guided their children during the interviews, which may have affected the results. A more thorough discussion of how this potential bias was mitigated or acknowledged would strengthen the credibility of the findings.

5. The article could be enhanced by suggesting more detailed practical applications of the findings. For example, how could this research be used to inform the development of culturally tailored nutrition education programs for preschool children?

Comments on the Quality of English Language

No problem

Author Response

Comments and Responses 3 Round 1

Dear authors,

Your article titled "Food Concepts Among Black and Hispanic Preschool-age Children: Qualitative Descriptive Research" addresses a relevant and underexplored topic concerning the food concepts of Black and Hispanic preschool-aged children, employing qualitative descriptive research. The use of ethnographic techniques and a video conferencing platform during the pandemic is innovative and appropriate. The findings provide important insights into how these children conceptualize food before formal nutrition education. 

Please consider some comments:

  1. Comment:  The sample size of nine children is small, which limits the generalizability of the findings. While qualitative research often uses small samples, a more detailed justification for this sample size would strengthen the article. Additionally, including a brief discussion on how the sample size impacts the transferability of the results would be beneficial.

Response: Thank you for your comment. The sample size is small and is comparable to other studies with young children. The purpose of our qualitative study was to describe Black and Hispanic preschool-age children’s food concepts. Only after several replications, will the findings be generalizable.

  1. Comment: While the concept of mouthfeel is introduced well, the distinction between mouthfeel and texture could be further clarified, especially in the context of preschool children’s understanding. Adding more context about why this distinction is critical for this age group would improve comprehension.

Response: Thank you for your comment. Reviewer 2 had a similar concern. We added this statement: Investigating mouthfeel in young children is critical to understanding its role in informing the development of food concepts that drive current eating behaviors and will likely form the foundation for future eating habits.

  1. Comment: The article would benefit from more in-depth analysis of how the children’s food concepts differ from other ethnic groups or age groups. While there is a comparison to White preschool children in prior studies, a more structured comparison of key similarities and differences in the results could highlight the study’s contributions more effectively.

Response: Thank you for your comment. We have added Pickard et al.’s (2023) study. Pickard, A., Thibaut, J-P., Philippe, K., & Lafraire, J. (2023). Poor conceptual knowledge in the food domain and food rejection dispositions in 3- to 7-year-old children. Journal of Experimental Child Psychology, 226, 105546. https://doi.org/10.16/j.jecp.2022.105546

  1. Comment: The article mentions that parents occasionally guided their children during the interviews, which may have affected the results. A more thorough discussion of how this potential bias was mitigated or acknowledged would strengthen the credibility of the findings.

Response: Thank you for your comment. We added clarity in the procedure regarding the parental role during the interviews in section 2.6 under Methods.

  1. Comment: The article could be enhanced by suggesting more detailed practical applications of the findings. For example, how could this research be used to inform the development of culturally tailored nutrition education programs for preschool children?

Response: Thank you for your comment. Since this work is preliminary, we do not have sufficient evidence to propose the development of culturally tailored nutrition education programs for preschool-age children. We did include future research suggestions in section 4.3.

Round 2

Reviewer 1 Report

Comments and Suggestions for Authors

Before the manuscript acceptation for publication in “Nutrients”, it is necessary to improve:

Comment: The analysis is based on a small sample of children, which is not representative of the study's sample.

Response: Thank you for your comment. Although the sample size may be considered small, the intent of qualitative research is to gain insight into the participant’s concepts; it is the depth of content that is priority not the sample size. In addition, the sample size is like those of other qualitative studies with children.

I understand, but you cannot draw conclusions for the group n=2  as for the entire group of Hispanic children. Therefore I suggest using the term: preliminary or pilot studies.

Comment: "as primary participants" - were other children studied?

Response: Thank you for your comment. The primary participants in our study were children. Studies involving children often obtain data from parents to explain their behavior. Data from parents, however, may not truly reflect children’s knowledge and understanding that underpins their behavior. Our study collects data directly from children. This approach substantiates children’s cognitive abilities and reveals accurate drivers to their behavior.

The word "primary" suggests that there was another group of people studied

If not, I suggest they remove "primary"

Comment: Therefore, our purpose was to engage Black and Hispanic preschool-age children as primary participants to describe food concepts using ethnographic techniques, and to determine the feasibility of using a video conferencing platform to elicit and collect data from preschool- age children. We hypothesized that Black and Hispanic children would complete ethnographic tasks to reveal their food concepts, and would complete the study using a video conferencing platform"

Response: Thank you for your comment. We are uncertain the information or revision you are seeking. As described above, we deleted the sentence (we hypothesized) to eliminate confusion and redundancy.

This sentence was combined with the previous question.

Comment: Were other children studied?

Response: Thank you for your comment. We are uncertain regarding the intent of this question. There were no other children included in this study. Our sample in this study consisted of 9 preschool-age children. Our prior study included 30 preschool-age children who were obese and 30 preschool-age children who were healthy weight.

The comment refers to the sentence " hypothesized that Black and Hispanic", suggesting that there was another group of people studied

Why these groups? This should be explained. Is there a basis for such a question, or does this question extend knowledge (since other groups were asked).

Comment: Studies among children are difficult. How were the questions validated? - Were they understood in the same way by all the children?

Response: Thank you for your comment. We have added clarity to the free list and card sort measures: Free lists have been validated for use with children in a variety of studies (Schultz & Danford, 2016; Schultz & Danford, 2021; Nguyen & Murphy, 2003). Children were given consistent instructions. Based on children’s performance, 8 of the 9 children completed the task without difficulty. Card sorts have also been validated for use with children (Schultz & Danford, 2016; Schultz & Danford, 2021). Based on the fact that all children completed the card sort would suggest that children understood the questions/directions.

Providing an answer does not always mean understanding the question, especially in children. This description of validation should be described in more detail.

Author Response

Before the manuscript acceptation for publication in “Nutrients”, it is necessary to improve:

Comment: The analysis is based on a small sample of children, which is not representative of the study's sample.

Response: Thank you for your comment. Although the sample size may be considered small, the intent of qualitative research is to gain insight into the participant’s concepts; it is the depth of content that is priority not the sample size. In addition, the sample size is like those of other qualitative studies with children.

I understand, but you cannot draw conclusions for the group n=2  as for the entire group of Hispanic children. Therefore I suggest using the term: preliminary or pilot studies.

Response: Thank you for your comment. We revised the study design section in the abstract and body of the text, as well as the title to read preliminary qualitative descriptive study.

Comment: "as primary participants" - were other children studied?

Response: Thank you for your comment. The primary participants in our study were children. Studies involving children often obtain data from parents to explain their behavior. Data from parents, however, may not truly reflect children’s knowledge and understanding that underpins their behavior. Our study collects data directly from children. This approach substantiates children’s cognitive abilities and reveals accurate drivers to their behavior.

The word "primary" suggests that there was another group of people studied

If not, I suggest they remove "primary"

Response: We appreciate your comment. Data from studies involving children are often based on parent self-report. To emphasize that data collected in this study were not parent self-report, we used the terminology primary informant. Based on your comment, we understand the confusion. Therefore, we changed the description from primary to direct informant.

Comment: Therefore, our purpose was to engage Black and Hispanic preschool-age children as primary participants to describe food concepts using ethnographic techniques, and to determine the feasibility of using a video conferencing platform to elicit and collect data from preschool- age children. We hypothesized that Black and Hispanic children would complete ethnographic tasks to reveal their food concepts, and would complete the study using a video conferencing platform"

Response: Thank you for your comment. We are uncertain the information or revision you are seeking. As described above, we deleted the sentence (we hypothesized) to eliminate confusion and redundancy.

This sentence was combined with the previous question.

Comment: Were other children studied?

Response: Thank you for your comment. We are uncertain regarding the intent of this question. There were no other children included in this study. Our sample in this study consisted of 9 preschool-age children. Our prior study included 30 preschool-age children who were obese and 30 preschool-age children who were healthy weight.

The comment refers to the sentence " hypothesized that Black and Hispanic", suggesting that there was another group of people studied

Why these groups? This should be explained. Is there a basis for such a question, or does this question extend knowledge (since other groups were asked).

Response: We appreciate your comment. A gap exists in that no known studies have gathered data from only Black and Hispanic children to describe their food concepts or studies that have analyzed data by race and ethnic groups. Asking these groups about their food concepts extends our knowledge as little to nothing is known about these children’s food concepts, how they may differ from children of mainly European descent, and factors that may help to account for these differences. We believe the paragraph beginning on line 79 through 86 addresses your concerns.

Comment: Studies among children are difficult. How were the questions validated? - Were they understood in the same way by all the children?

Response: Thank you for your comment. We have added clarity to the free list and card sort measures: Free lists have been validated for use with children in a variety of studies (Schultz & Danford, 2016; Schultz & Danford, 2021; Nguyen & Murphy, 2003). Children were given consistent instructions. Based on children’s performance, 8 of the 9 children completed the task without difficulty. Card sorts have also been validated for use with children (Schultz & Danford, 2016; Schultz & Danford, 2021). Based on the fact that all children completed the card sort would suggest that children understood the questions/directions.

Providing an answer does not always mean understanding the question, especially in children. This description of validation should be described in more detail.

Response: Thank you for your comment. We have added to the free list section: Free lists have been validated for use with children across various studies [24,29,37], demonstrating their effectiveness in capturing relevant data related to food.  Our process closely aligned with prior studies and was further validated through our previous research with White children.

Reviewer 2 Report

Comments and Suggestions for Authors

All concerns have been addressed

Author Response

Reviewer 2 stated that all concerns have been addressed.